# Concurrence of High Corrosion Resistance and Strength with Excellent Ductility in Ultrafine-Grained Mg-3Y Alloy

**DOI:** 10.3390/ma15217571

**Published:** 2022-10-28

**Authors:** Mária Zemková, Peter Minárik, Eva Jablonská, Jozef Veselý, Jan Bohlen, Jiří Kubásek, Jan Lipov, Tomáš Ruml, Vojtěch Havlas, Robert Král

**Affiliations:** 1Faculty of Mathematics and Physics, Charles University, Ke Karlovu 5, 121 16 Praha, Czech Republic; 2Research Centre, University of Žilina, Univerzitná 8215/1, 01026 Žilina, Slovakia; 3Faculty of Food and Biochemical Technology, University of Chemistry and Technology, Technická 5, 166 28 Prague, Czech Republic; 4Helmholtz-Zentrum Hereon, Institute of Material and Process Design, Max-Planck-Straße 1, 21502 Geesthacht, Germany; 5Faculty of Chemical Technology, University of Chemistry and Technology, Technická 5, 166 28 Prague, Czech Republic; 6Second Faculty of Medicine, Charles University, V Úvalu 84, 150 06 Praha, Czech Republic

**Keywords:** magnesium, yttrium, ultrafine-grain microstructure, corrosion rate, mechanical properties

## Abstract

In the field of magnesium-based degradable implantable devices, the Mg-Y-RE-Zr alloying system (WE-type) has gained popularity due to its satisfying degradation rate together with mechanical strength. However, utilization of RE and Zr in the WE-type alloys was originally driven to improve Mg-based alloys for high-temperature applications in the industry, while for medical purposes, there is a question of whether the amount of alloying elements may be further optimized. For this reason, our paper presents the Mg-3Y (W3) magnesium alloy as an alternative to the WE43 alloy. This study shows that the omission of RE and Zr elements did not compromise the corrosion resistance and the degradation rate of the W3 alloy when compared with the WE43 alloy; appropriate biocompatibility was preserved as well. It was shown that the decrease in the mechanical strength caused by the omission of RE and Zr from the WE43 alloy could be compensated for by severe plastic deformation, as achieved in this study, by equal channel angular pressing. Ultrafine-grained W3 alloy exhibited compression yield strength of 362 ± 6 MPa and plastic deformation at maximum stress of 18 ± 1%. Overall, the early results of this study put forward the motion of avoiding RE elements and Zr in magnesium alloy as a suitable material for biodegradable applications and showed that solo alloying of yttrium is sufficient for maintaining desirable properties of the material at once.

## 1. Introduction

In recent years, there has been a growing effort to use an intentionally corrodible material in various critical tissue engineering and regenerative therapies [1]. Examples of such applications include cardiovascular stents or temporary orthopaedic fixation devices that are required in the body only for the specific time of healing. These materials are characterized by requirements for specific mechanical properties and corrosion rates. For example, according to Erinc et. al. [2], the corrosion rate of bone fixtures needs to be less than 0.29 mg/cm^2^/day in simulated body fluid at 37 °C, and the strength needs to be higher than 200 MPa. Metal-based materials are considered to be more suitable for load-bearing applications in comparison with degradable polymers, especially because of their superior mechanical strength. However, it is important to keep in mind that released metallic ions may cause physiological responses in the surrounding tissue or inner organs. It was shown that magnesium ions released through corrosion do not cause any local or systematic toxicity [3]; on the contrary, magnesium has an osteoconductive effect in the surrounding tissue [4]. However, pure Mg is soft and does not provide sufficient strength—its tensile strength is only ~20 MPa [5]. Thus, Mg is commonly combined with various alloying elements in order to improve not only mechanical properties but also corrosion and wear resistance. In general, the corrosion of a metal in its environment is a complex system, and both internal (material) and external (environment) parameters can play a significant role. Regarding Mg-based alloys, many studies have shown that the main factors in controlling corrosion rate are chemical composition in combination with microstructure [3,4,6,7,8,9,10,11].

Yet most of the currently investigated magnesium-based alloys, such as AZ31 [6,7], AZ91 [6,8], ZE41 [9,10], and WE43 [9,11], which are studied for medical applications, were originally designed for automotive and aerospace industries without considering biocompatibility [3]. The most often investigated alloy in various in vitro or in vivo studies is Mg-Y-RE-Zr containing 4 wt% yttrium, 3 wt% mischmetal, and ~0.4 wt% zirconium [6,9,12]. Mischmetal is an unspecified combination of rare earth (RE) elements that are not highly toxic [13,14]. Nevertheless, from the biological point of view, utilization of mischmetal with unspecified composition may be potentially hazardous. Thereby, the major question is whether it is necessary to use the RE elements for mechanical profile purposes or whether alternatives are available. The omission of Zr seems to be valid as well, since it was found that an already low concentration of Zr in solid solution in Mg acts as an anodic activator and increases the corrosion rate of alloy [15].

It is generally known that the addition of mischmetal and yttrium remarkably improves mechanical properties due to the solution and precipitation hardening [16]. However, the excellent corrosion resistance of this alloy is obtained primarily by the yttrium addition [12] and the subsequent formation of the Y_2_O_3_ surface layer, which slows down the corrosion rate [17,18,19]. During casting, zirconium is added to the commercial WE-type alloys to stimulate grain nucleation [20], which causes the formation of a significantly finer grain structure in the cast billets. Accordingly, omitting zirconium and mischmetal could have had a detrimental effect on the mechanical properties of the materials. Nevertheless, this phenomenon can possibly be compensated by a significant change in the microstructure. For this reason, equal channel angular pressing (ECAP) was used in this work. This technique is a very effective method for the refinement of the coarse-grained microstructure and increasing the strength of alloy by grain boundary hardening. Moreover, the utilization of ECAP may be very useful for manufacturing bulk metallic implants because of its scalability and effectiveness [21,22,23].

Thus, this study was focused on investigating the Mg-Y alloy processed by ECAP as a potential alloy for biodegradable applications. The effect of microstructural changes on the in vitro degradation in biological media, cytotoxicity, and mechanical properties were investigated. Commercial Mg-Y-RE-Zr (WE43) alloy was processed similarly and was used as a benchmark in all tests. Because we intend to compare our results with the well-known system Mg-Y-RE-Zr, which usually contains 4 wt% Y (WE43 alloy), we decided to maintain the Y amount close to the WE43 alloy and try to separate the effect of the Y and the effect of the other alloying elements. Nevertheless, according to the previous reports, we intentionally decreased the yttrium content from 4 wt% to 3 wt% because it was shown that such a decrease could improve both mechanical and corrosion properties [18,19].

## 2. Material and Methods

### 2.1. Experimental Material

Two magnesium alloys were used in this study—the commercial alloy WE43 (Mg-4Y-3RE) and the experimental alloy W3 (Mg-3Y). The actual composition measured by spark emission spectroscopy (Berwyn, PA, USA) is shown for both alloys in Table 1. The WE43 alloy was supplied in the as-cast condition, and the W3 alloy was cast using a conventional casting process from pure elements. Both materials were homogenized by thermal treatment at 525 °C (WE43) and 400 °C (W3) for 16 h. The preliminary study showed that the grain size of the cast WE43 alloy was fine enough to be subsequently processed by ECAP (~150 µm), while the microstructure of the W3 alloy was formed by grains exceeding 2 mm in diameter. Therefore, the W3 alloy was extruded prior to the ECAP for initial grain refinement. The extrusion was performed at 350 °C with an extrusion ratio of 17 and a ram speed of 1 mm/s. 

The ultrafine-grained condition of both alloys was prepared by ECAP (Charles University, Prague, Czech Republic). Rectangular bars (1 cm × 1 cm × 10 cm) were machined and processed via route B_C_ up to 8 passes through the ECAP die. The angle between two intersecting channels and the corner angle of the ECAP die were 90° and 0°, respectively. The processing parameters, namely temperature and rate, are shown in Table 2.

### 2.2. Microstructure Analysis

The microstructure of both alloys was investigated using scanning electron microscope (SEM) FEI Quanta 200F (Hillsboro, OR, USA), equipped with EDAX energy-dispersive X-ray spectroscope (EDS) and EDAX electron backscatter diffraction (EBSD) camera (EDAX, Pleasanton, CA, USA), and transmission electron microscope (TEM) Jeol 2200FS (Tokyo, Japan) equipped with Gatan EDS. Samples for SEM were prepared by mechanical polishing with decreasing particle size down to 50 nm. Samples prepared for EBSD were subsequently ion-polished using Leica EM RES102 (Wetzlar, Germany) ion-beam milling system. Samples for TEM were mechanically thinned and subsequently electrochemically polished using Struers Tenupol 5 (Cleveland, OH, USA) in a solution of perchloric acid and methanol. The crystallographic texture was measured by X-ray diffractometer (XRD) PANalytical XPert MRD (Malvern, UK). Cu_K_ radiation and polycapillary in the primary beam were employed during the measurements. Pole figures from 6 reflections, (0002), $10\bar{1}0$, ($10\bar{1}1$), ($10\bar{12}$), ($10\bar{1}3$), and ($11\bar{2}0$), were measured. Inverse pole figures were calculated using the MTEX toolbox implemented in the Matlab 2020a software [24].

### 2.3. Corrosion Rate

Corrosion performance in biological media was investigated in the minimum essential medium (MEM, Sigma M0446) with 10% of fetal bovine serum (FBS) for 14 days. This medium was chosen as a simulated body fluid since it reflects the composition of human plasma in terms of the concentration of most of the ions, and the presence of glucose and proteins. Samples with dimensions of 6 mm × 6 mm × 1 mm were cut from all investigated alloys. Just before the tests, the samples were ground (SiC1200, 15 μm) to remove the naturally occurring corrosion layer. Subsequently, the initial weight (*m*_0_) of each sample was documented and the sterilized samples (2 h in ethanol and 2 h under UV light) were immersed into 40 mL of media in falcon tubes with vented caps (Corning, New York, NY, USA) and incubated in a 5% CO_2_ and humidified atmosphere at 37 °C on an orbital shaker. pH was stabilized using a bicarbonate buffer system. Due to high relative humidity, evaporation was negligible. The medium was not exchanged during the test. After the immersion, the samples were rinsed with ethanol. Three replicates of each sample type were cleaned in order to remove the corrosion products according to the ISO 8407:2009 standard [25]. Finally, the weight of the samples (*m*) was measured again, and the corrosion rate was calculated as *C_r_ = (m*_0_ − *m)/(t* × *A)*, where *t* is the immersion time, and *A* is the total surface area of a sample.

### 2.4. In Vitro Cytocompatibility Tests on Extracts (Elution Test)

In vitro cytotoxicity on extracts (elution test) was performed in accordance with ISO 10993-5 standard. The metabolic activity was evaluated using the resazurin test. The samples (6 mm × 6 mm × 1 mm) were ground (SiC1200, 15 μm) and sterilized (2 h in 70% ethanol followed by UV irradiation for 2 h). Afterwards, the samples were shaken (125 rpm) in MEM + 5% FBS at 37 °C in closed containers for 24 h. A lower concentration of serum in extraction media was used in accordance with the standard. The amount of 1 mL of medium was used for 17 mm^2^ of the sample surface. This surface-to-volume ratio was five times higher than mentioned in ISO standard, which is recommended for degradable metals [26]. L929 murine fibroblasts (ATCC^®^ CCL-1™) were cultured under standard conditions in MEM + 10% FBS. The cells were seeded in 96-well plates in MEM + 5% FBS at a density of 1 × 10^4^ or 0.3 × 10^4^ cells per well, depending on the following incubation period. After 24 h, the medium was replaced by the extracts of the samples. MEM + 5% FBS was used as a negative control. After 24 or 72 h of incubation with the extracts, cell metabolic activity was determined using resazurin assay. The cells were first washed with phosphate-buffered saline (PBS) and resazurin (final concentration 25 µg/mL) in medium (MEM + 10% FBS without phenol red) was then added to each well. Fluorescence was measured after 1 h of incubation (ex/em wavelength being 560/590 nm). The percentage of metabolic activity of the cells exposed to extracts relative to the negative (untreated) control was evaluated. The extracts were prepared from at least two samples, and six iterations and extracts were performed.

### 2.5. Direct In Vitro Cytocompatibility Tests (Metabolic Activity)

The samples (6 mm × 6 mm × 1 mm) after the elution test (i.e., after 24 h pre-incubation in cultivation medium) were placed into a cultivation 12-well plate. The L929 cells were resuspended in the medium and seeded directly onto the pre-incubated samples. The test was performed in accordance with ISO 10993-5 standard. The volume was 2 mL, and the seeding density was 24,000 cells/cm^2^. After 24 h, the samples were removed and the metabolic activity of the cells growing in contact with the samples was measured as described above.

### 2.6. Contact In Vitro Cytocompatibility Tests (Microscopy)

The measured samples (5 mm × 5 mm × 1 mm) and the Ti-6Al-4V control samples (10 mm in diameter, 1 mm height) were ground (SiC4000, 5 µm) and sterilized (2 h in 70% ethanol followed by UV irradiation for 2 h). Ti-6Al-4V was used as a metallic cytocompatible material to control the process of cell preparation. Subsequently, the samples were subjected to 24 h pre-incubation in 2 mL of cultivation medium at 37 °C and 5% CO_2_. Human osteosarcoma cells U-2 OS (ATCC^®^ HTB-96™) were used for the cytocompatibility testing due to their resemblance to osteoblasts. U-2 OS were cultivated in DMEM (Sigma D0819) supplemented with 10% FBS. The cells were resuspended in the medium and seeded directly onto the pre-incubated samples placed in a 12-well plate. The volume was 2 mL, and the seeding density was 24,000 cells/cm^2^. After 24 h, the cells growing on the samples were examined using fluorescence and electron microscopes. For fluorescence microscope, samples were rinsed with PBS and fixed with 4% formaldehyde for 20 min at room temperature. Afterwards, cell membranes were permeabilized with Tween-20 (0.1%, 15 min), F-actin was visualized after the incubation with phalloidin-TRITC (0.5 µg/mL, 15 min), and nuclei were stained with DAPI (0.5 µg/mL, 5 min). Samples were then observed using an inverted microscope Olympus IX81 (Hamburg, Germany). For SEM, the samples were rinsed with PBS and fixed with Karnovsky’s fixative (2% formaldehyde, 2.5% glutaraldehyde, and 2.5% sucrose in 0.2 M cacodylate buffer) for 1.5 h. Samples were then rinsed with 0.1 M cacodylate buffer and dehydrated in a series of ethanol (50%, 70%, 90%, and 100%, 10 min). Subsequently, the samples were transferred to acetone and dried using Leica EM CPD300 (Wetzlar, Germany). Finally, the samples were sputter-coated with gold (10 nm).

### 2.7. Mechanical Tests

Uniaxial compression tests were performed by Instron 5880 (Norwood, MD, USA) at room temperature with an initial strain rate of 10^−3^ s^−1^. Four rectangular samples having dimensions of 4 mm × 4 mm × 6.5 mm were cut from each investigated material with the deformation direction parallel to the ECAP/extrusion processing direction. Additionally, the Vickers microhardness tests were performed by fully automatic testing device QNESS Q10a (ATM Qness, Mammelzen, Germany). Vickers indenter applied load of 0.1 kg for 10 s for all measured circles. At least 200 indents were measured for each investigated sample. The microhardness tests were quantified from the planes perpendicular to the ECAP/extrusion processing direction.

## 3. Results

### 3.1. Microstructure Characterization

The microstructure of the coarse-grained and ultrafine-grained conditions of Mg-Y (W3) and Mg-Y-RE-Zr (WE43) alloys was primarily analyzed by SEM and TEM, including EBSD or (transmission Kikuchi diffraction) TKD. Figure 1a,c show the grain structure of the coarse-grained condition of both investigated alloys before processing through ECAP. Both samples in the initial condition exhibited fully recrystallized microstructure with equiaxed grains. The average grain size of the WE43-0P sample was ~150 µm, while it was only ~35 µm in the case of the W3-0P sample. Note that the W3 alloy needed to be extruded prior to ECAP because of the presence of too-large grains in the as-cast condition. The analysis also showed that the solid solution treatment of both alloys led to the complete dissolution of secondary phase particles in the case of the WE43 alloy, and only a sporadic presence of secondary phase particles was observed in the W3 alloy. Extrusion performed on the W3 alloy did not cause precipitation of the secondary phases because it was performed above the solvus, and the material was rapidly cooled down. Only a small amount of secondary phase particles was present in the material, and they were aligned in the stripes along the extrusion direction (not shown here). TEM investigation revealed that the particles are Mg_24_Y_5_ phase; for details, see Ref. [27].

The subsequent processing by ECAP led to significant grain refinement of both investigated alloys. The final average grain size was ~350 nm and ~730 nm for WE43-8P and W3-8P samples, respectively. In addition to the difference in the degree of grain refinement, there was a distinct difference in the precipitation behaviour during the processing between the two investigated alloys. The WE43 alloy was processed below the solvus, and repeating precipitation and dissolution of the secondary phase particles during each pass led to the formation of fine and almost spherical particles. Finally, the WE43-8P sample exhibited a uniform and dense distribution of secondary phase particles having the size of ~150 nm, see Figure 2a. TEM investigation revealed that these particles are Mg_5_RE phase by selected area electron diffraction (SAED), see Figure 2b. For the detailed description of the microstructure evolution of the WE43 alloy, the reader is referred to Ref. [28]. On the other hand, the processing of W3 alloy was performed well above the solvus temperature; therefore, an almost negligible amount of secondary phase particles in the W3-8P sample was found (Figure 2c). The particles were identified as Mg_24_Y_5_ phase in TEM by SAED (Figure 2d). The suppressed precipitation means that almost all yttrium was dissolved in the magnesium matrix. In addition, there was a significantly higher amount of residual strain in the W3-8P sample after the processing than in the WE43-8P sample: note the huge amount of dislocation tangles in Figure 2c. 

### 3.2. Corrosion in Biological Media

The corrosion behaviour of the investigated samples was further examined by means of the immersion tests and the calculation of corresponding corrosion rates. The corrosion rate of all studied samples was measured in two types of biological media. MEM, which is the standard medium used in literature, and MEM + 10% FBS, which reflects the composition of human plasma in terms of the presence of glucose and proteins and in terms of the concentration of the ions. The samples were immersed in both solutions for 14 days, and subsequently, the corrosion rate was calculated. The resulting values are shown in Figure 3. The type of media was found to strongly influence the overall corrosion rate of studied samples. Additionally, the result of this measurement shows that while in the MEM, there is a statistically significant difference in the corrosion rate of the individual conditions, and there is literally no difference in the MEM + FBS media. Surprisingly, for both ultrafine-grained conditions immersed in MEM media, a higher corrosion rate was observed compared with their coarse-grained counterparts. Nevertheless, the presence of FBS (more realistic simulated body fluid) caused a decrease in the corrosion rate of both studied alloys, while it did not affect the results of the WE43-0P sample within the calculated error.

According to measured data in MEM + 10% FBS, the corrosion rate of the W3-8P was 0.29 mg/cm^2^/day, while in the case of the WE43-8P, it was 0.31 mg/cm^2^/day. Within the statistical error, no difference in corrosion rate between ECAP conditions of the W3 and WE43 alloys was observed. As was mentioned above, the upper limit for the corrosion rate of temporary bone fixation implants is 0.29 mg/cm^2^/day in the simulated body fluid. The measured data unambiguously showed that W3 alloy exhibited satisfactory performance for that type of application. Note that the pH value of both media was ~7.5 during the entire immersion period.

### 3.3. In Vitro Cytocompatibility Tests

Cytotoxicity of all studied conditions was thoroughly investigated by indirect (elution) and also direct tests using L929 murine fibroblasts. The elution test was conducted with two incubation periods—24 h and 72 h. Subsequently, the viability measured as the metabolic activity of the cells was analyzed, and the resulting relative values of the metabolic activity with respect to the negative control (sole extraction medium) are shown in Figure 4. It is shown that the metabolic activity of the cells was not changed by omitting mischmetal and zirconium, nor by the ECAP processing. The results were unaffected by the prolonged incubation period from 24 h to 72 h. Cell viability of all samples was far above the 70% limit stated in ISO 10993-5 standard. 

The direct cytocompatibility tests are considered to be stricter than the indirect ones. The test was performed by a cultivation of the L929 cells directly with the investigated samples for 24 h. The result of the subsequently measured metabolic activity of the exposed cells is shown in Figure 5. As is clearly shown, the results are comparable to the ones acquired from the elution test. The metabolic activity was far above the 70% normative limit; therefore, all investigated materials can be considered non-cytotoxic also according to this stricter test. 

Finally, the U-2 OS human osteosarcoma cells were cultivated directly on the surface of the investigated samples for 24 h, and subsequently, the samples were analyzed by fluorescence microscope and SEM. The resulting micrographs are shown in Figure 6. The investigation showed that the cells colonized all types of samples. The cells were well spread with developed actin fibres. Also, evident amitosis is visible in the micrographs from the fluorescence microscope. Note that only the results of the ultrafine-grained samples of both investigated alloys are shown because the results were qualitatively similar to their coarse-grained counterparts.

### 3.4. Analysis of the Corrosion Layer Morphology and Composition

Due to the better understanding of the corrosion performance of studied alloys in MEM biological media after 14 days of immersion, the corrosion layers of all four samples were examined by SEM. Note that the MEM solution was selected for this analysis because differences in the degradation rate were observed between the alloys and the type of samples. In the case of MEM + 10% FBS solution, all samples exhibited a comparable degradation rate. The cross-sectional cut through the developed corrosion layer is shown for all samples in Figure 7a. The darkest area in each figure corresponds to the epoxy resin used for the sample fixation during its preparation for SEM, the dark grey area accounts for the corrosion layer formed on the sample surface, and the light grey zone corresponds to the sample. The figures clearly depict that the corrosion front propagated uniformly in all samples except the W3-8P, where the formation of partially connected pockets was observed. Nevertheless, the corrosion layer thickness in the W3-8P sample was, on average, lower than its coarse-grained counterpart (W3-0P). It is also visible that the corrosion layer consists of profound cracks in all samples. Not surprisingly, the WE43-0P exhibited the thinnest corrosion layer (~20 μm), almost four times lower compared with the rest of the samples. The thin corrosion layer could indicate suppression of the corrosion front propagation, which corresponds to the results of the corrosion resistance and the mass loss tests in Section 3.2 and Section 3.3, respectively. On the contrary, the investigation carried out on the surface of the sample showed that there was significant delamination of the corrosion layer in the WE43-0P sample (see Figure 7b), which was not observed in any other one, cf. Figure 7c, and could distort the measurement of the corrosion layer thickness.

In general, Mg degrades in the aqueous environment and produces a corrosion layer mainly composed of magnesium oxide MgO and/or magnesium hydroxide Mg(OH)_2_ [3,9,11,29] on the fresh surface. Elemental mapping across the corrosion layer of 8P conditions of both alloys (see Figure 8) showed that yttrium (neodymium) is being uniformly built in the corrosion layer of both alloys. Increased content of calcium and phosphor in the outer part of the surface layer proves the formation of stable amorphous calcium phosphates (CaP) [30,31], which commonly form during the degradation of magnesium alloys in biological media. Generally, the composition of the corrosion products depends directly on that of the immersion media.

### 3.5. Mechanical Properties

Mechanical properties of all investigated samples were studied by standard compression deformation tests at room temperature and Vickers microhardness tests. The true plastic stress–strain curves are presented in Figure 9. The evaluated values of the compressive yield strength (CYS), ultimate compression strength (UCS), and deformation at maximum stress are presented in Table 3. The comparable values of the CYS and UCS were observed for the initial condition of both alloys regardless of the different grain sizes. After ECAP processing, the CYS increased significantly in both alloys. The CYS value of the WE43-8P was substantially higher than that of the W3-8P alloy, while the UCS values remained comparable. On the other hand, the W3 alloy exhibited significantly (double) higher deformation at maximum stress values in both coarse-grained and ultrafine-grained conditions. ECAP processing also resulted in a significant change in the deformation curve shape. The coarse-grained samples exhibited a typical power-law shape (WE43) and weak S-shape (W3), while a distinct sharp yield point was present in the deformation curve of both ultrafine-grained materials, see Figure 9. In addition, the Vickers microhardness testing (HV_0.1_) were performed on the cross-section of all the investigated samples, and the average values of the microhardness are included in Table 3. Similar trends to those of the CYS/UCS were also observed in the case of the HV_0.1_ values, which increased clearly and steadily from 73 ± 5 or 57 ± 3 (WE43-0P or W3-0P, respectively) to 114 ± 3 or 100 ± 3 (WE43-8P or W3-8P, respectively).

### 3.6. Texture

In order to fully explain differences in the deformation behaviour of the investigated samples, XRD texture measurement was performed. The inverse pole figures (IPF) corresponding to the W3-0P, W3-8P, and WE43-8P samples calculated for the extrusion/ECAP processing direction are shown in Figure 10. Note that texture for the WE43-0P sample was not measured because it is a cast condition and is considered to be texture-free. All three samples exhibited weak texture with the typical texture components for the individual processing steps. IPF of the W3-0P sample showed a distinct peak in the $\left[10\bar{1}0\right]$ corner, which corresponds to the formation of $\left\{10\bar{1}0\right\}$ fibre during the extrusion. This fibre usually forms during the extrusion of the magnesium alloys because of the predominant activation of the basal slip system [32]. The less-developed texture element occurring in the middle of the IPF triangle corresponds to grains that have their c-axis tilted from the extrusion direction by ~50°. This texture element usually forms in magnesium alloys containing Y and/or RE [33]. ECAP processing resulted in the formation of two texture components in the W3-8P sample and only one in the WE43-8P sample. It was repeatedly shown that the texture component denoted as “a” is formed by the predominant activation of the basal slip system [33,34,35], and the component denoted as “b” is formed by the predominant activation of the prismatic and <c + a> pyramidal slip system [36,37,38]. It is important to note that the “b” component was stronger in the W3-8P sample, while the “a” component was missing in the WE43-8P sample.

## 4. Discussion

The effect of ECAP on the microstructure changes, in particular grain refinement, was investigated already in many studies using a variety of magnesium alloys; see, for example, [7,22,28,36]. In this work, ECAP was primarily used to prepare ultrafine-grained conditions of both investigated alloys and compensated for the possible detrimental effect on the mechanical properties due to the omission of Zr and RE from WE43 alloy. ECAP processing parameters were derived from the previous study performed on the WE43 alloy and led to comparable grain size (~350 nm) [28]. Even so, a slightly coarser grain structure was observed in the W3 alloy after the processing (~730 nm). In the WE43 alloy, rare earth elements formed small precipitates that increased the effectivity of the grain refinement through dynamic recrystallization and suppressed grain growth [39]. Consequently, small equiaxed grains formed without significant residual strain. On the other hand, there was almost no precipitation occurring during the processing of the W3 alloy. Yttrium dissolved in the magnesium matrix facilitates non-basal slip systems [27], and consequently, grain refinement was more effective than in pure magnesium [40]. In addition, Hadorn et al. [41] showed that Y tends to segregate to grain boundaries and can effectively hinder their mobility during dynamic recrystallization. The high activity of various slip systems is also responsible for the significant residual strain observed in the W3-8P condition, despite the relatively high processing temperature.

The most important results of this study lay in a comparison of the degradation rate and cytotoxicity of the investigated alloys. Omitting mischmetal, which is an unspecified combination of rare earth elements, and zirconium did not cause a significant change in the degradation rate of the W3 alloy compared with the WE43 alloy. According to the previous comparative study carried out in MEM on the number of binary alloys, including Mg-Y, alloying with sole yttrium causes an increase in the degradation rate with increasing content and leads to corrosion localization [10]. However, a much lower degradation rate was observed in this study regardless of higher yttrium content (3wt% compared with 0.5 wt% and 2 wt% used in Ref. [10]), and no localization occurred. Therefore, the discrepancy in these results can be ascribed to the microstructural condition of the alloys, which were not specified in Ref. [10] and cannot be discussed. Conversely, studies that investigated Mg-Y-RE systems showed superior corrosion resistance for the WE43 alloy, which exhibited the lowest corrosion rate of 0.15 mg/cm^2^/day measured in SBF after exposure for 7 days [42] (SBF biological media is similar to MEM solution, but free to amino acid, vitamins, and glucose). In our study, the corrosion rate was calculated from the immersion tests performed in two biological media, MEM and MEM + 10% FBS. The degradation rate in MEM was affected by the processing—a higher degradation rate was observed in samples with UFG microstructure. The negative effect of a higher amount of lattice defects, in this case grain boundaries, on the corrosion resistance was repeatedly shown in the literature [43]. However, the difference in the degradation rate between the investigated alloys was very small for the coarse-grained condition and within the error for the UFG condition. On the other hand, the same (within the error) corrosion rate of ~0.3 mg/cm^2^/day was observed in all samples immersed in MEM + 10% FBS regardless of the alloy composition or microstructure. Note that MEM + 10% FBS reflects the composition of human plasma more realistically and, therefore, these results predict in vivo degradation rate of the material more accurately. The observed overall decrease in the corrosion rate in the MEM + 10% FBS solution, compared with pure MEM, is a common effect of FBS mainly due to the presence of proteins [44]. Our results are in accordance with the previous ones that showed a decrease in the corrosion rate of pure magnesium [45] or magnesium alloy [46] in the presence of FBS. According to the literature, several theories have been proposed to describe the effect of the presence of proteins on corrosion rate: (i) proteins are able to change metallic corrosion by changing anodic or cathodic processes, (ii) proteins interact with corroded surface and create layer that further inhibits the corrosion, and (iii) proteins together with Ca and P form an insoluble salt layer which provide additional protection. Nevertheless, a previously observed effect of grain refinement on degradation rate in the FBS-containing solution [7,46] was not observed. The results of this study unambiguously proved that omitting mischmetal and zirconium does not affect the degradation rate of the W3 alloy, and this material is of very high interest in the development of biodegradable magnesium implants.

The effect of alloy composition and processing on the development of the corrosion layer was investigated only for samples exposed to MEM solution because in this solution, differences in the degradation rate were observed, while in MEM + 10% FBS solution, the degradation rate was the same within the data scatter. A significantly thinner layer of the corrosion products, observed in the WE43-0P sample (Figure 7a), is a result of massive fragmentation and delamination of the corrosion layer observed only in this sample, cf. Figure 7b,c. There are reports showing a decrease in fragmentation of the corrosion layer products developed on different magnesium alloys after grain refinement. A finer grain structure causes the formation of a more coherent scale [47] in which structural integrity may be stabilized by unsolvable oxides of the alloying elements [22,46]. Consequently, the WE43-0P sample with ~5× larger average grain size, compared with the W3-0P sample, was the only sample in which severe fragmentation of the corrosion layer occurred. The formation of pockets in the W3-8P alloy was caused by the residual strain resulting from ECAP and its uneven distribution, which is clearly seen in Figure 2c. Areas with higher internal strain act more anodically, and the corrosion front propagates faster [48,49]. Note that this effect does not cause too significant corrosion localization, which is able to lead to severe pitting. The assumption that the residual strain is responsible for the pocket formation is also supported by the EDS analysis shown in Figure 8. It is shown that the major alloying elements—yttrium in the W3 alloy and both yttrium and neodymium in the WE43 alloy—are uniformly distributed in the corrosion products. Severe corrosion localization usually occurs in samples with large secondary phase particles [17,18,46], which were not present in the studied materials.

A thorough investigation of cytotoxicity showed that the composition/microstructure of both alloys do not have a negative effect on cell viability. Both direct and indirect tests showed relative cell viability high above the 70% limit stated in ISO 10993-5 standard. In addition, the direct cultivation of the U-2 OS human osteosarcoma cells on the investigated samples did show that the cells successfully colonized all samples, the cells were well spread with developed actin fibres, and in some cells, amitosis occurred. Therefore, it is evident that the environmental conditions were favourable. This result is not surprising considering the repeatedly proven biocompatibility of the WE43 alloy [42,50]. Yet, it is important to show that omitting mischmetal and zirconium did not compromise biocompatibility, and the W3 alloy can be considered biocompatible as well.

In addition to the degradation properties and biocompatibility, the mechanical strength of the potential materials for medical applications is very important. As mentioned in the Introduction, the composition of the Mg-Y-RE-Zr alloys was initially tailored for applications in which the mechanical strength of the material had to sustain high temperatures. It was shown that omitting mischmetal and zirconium did not negatively affect the degradation rate and biocompatibility of the W3 alloy, but a negative effect on the room-temperature mechanical strength was expected. Table 3 shows that the CYS values of both coarse-grained samples are comparable, as well as values of HV_0.1_, but it must be taken into account that the grain size of the WE43-0P sample is almost five times higher. Therefore, a significant contribution of solid solution strengthening of mischmetal and zirconium is responsible for the comparable values of CYS and HV_0.1_. In addition, the effect of texture plays an important role in the W3-0P sample. The distinct fibre component formed during the extrusion (Figure 10) and is responsible for the massive activation of twinning in this sample, which manifested itself by a typical S-shape of the deformation curve (Figure 9). Because of the significantly lower concentration of the alloying elements in the W3 alloy, the strengthening potential comes primarily from the grain refinement. For this reason, ECAP processing was applied in this study. CYS values of both 8P samples were increased by more than 300% compared with their corresponding initial conditions. It was shown earlier that ECAP significantly refines the grain size and increases the strength of magnesium-based alloys such as AZ31 [23], LAE442 [36], and WE43 [28,51]. CYS of the W3-8P sample was 362 ± 6 MPa, which is significantly higher in commonly investigated WE43 alloy processed by various methods, including severe plastic deformation, as shown in Table 4. It is important to note that an increase in mechanical strength positively affects the minimum implant size needed to sustain the load. Nonetheless, the strengthening of the WE43 alloy by ECAP was even higher because of the presence of mischmetal and zirconium. TEM investigation revealed that, unlike in the W3 alloy, uniform distribution of fine intermetallic particles formed in the WE43-8P sample. In addition, “a” texture component (Figure 10) was missing in the WE43-8P sample, which is often responsible for texture softening [36]. The occurrence of the sharp yield point in both 8P samples is caused by the formation of the “b” texture component (Figure 10), as shown in Ref. [52].

Besides mechanical strength, deformability is a very important parameter for implant manufacturing material. Some implants, especially fixation plates, are being deformed during the surgery to fit the patient. Therefore, deformability cannot be sacrificed for strength. ECAP-processed materials shown in this study exhibited very yield strength, but the W3-8P sample is of particular interest because it combines superior strength compared with results shown in Table 4 while keeping relatively high deformation at maximum stress. Furthermore, this is still at a relatively low concentration of alloying elements compared with other developed materials.

## 5. Conclusions

This study focused on the investigation of binary alloy Mg-3Y (W3) processed by equal channel angular pressing (ECAP) as a potential new material suitable for biodegradable medical applications. The microstructure, mechanical properties, in vitro biocompatibility, and corrosion performance in biologic media were investigated and compared with well-known commercial Mg-Y-RE-Zr (WE43) magnesium alloy. The following conclusions may be drawn from this study:

The optimal parameters of ECAP processing resulted in significant grain refinement in both investigated alloys. The achieved average grain size was ~730 nm and ~350 nm for the W3 alloy and WE43 alloy, respectively. 

No evidence of cytotoxicity of the W3 alloy was revealed in the indirect test and two types of direct tests. Omitting the mischmetal and zirconium did not compromise biocompatibility, and the W3 alloy can be considered biocompatible, similarly to the WE43 alloy.

The convenient degradation rate of the WE43 alloy was not compromised by omitting the mischmetal and zirconium. The results showed that there is literally no difference in the corrosion rate in both investigated alloys performed in the minimum essential media + 10% fetal bovine serum solution. 

The mechanical strength of the W3 alloy processed by ECAP was more than 300% higher compared with the coarse-grained condition. The omitting of RE and Zr could be adequately compensated for by a significant grain refinement achieved by ECAP.

Mechanical properties showed that, despite decreased strength due to omitting the mischmetal and zirconium, the W3 has much higher deformability. This result is very important because implants such as plates are often deformed during surgery to fit the individual patient. Therefore, increased deformability extends the possibilities of future utilization of this material.

The results of this work showed that the ECAP-processed W3 alloy exhibits a good combination of ultrafine-grain grain structure, low crystallographic texture, high strength, and comparable degradation rate compared with the WE43 alloy. These attributes make this material very interesting for utilisation in medicine as biodegradable material.

## Figures and Tables

**Figure 1 materials-15-07571-f001:**
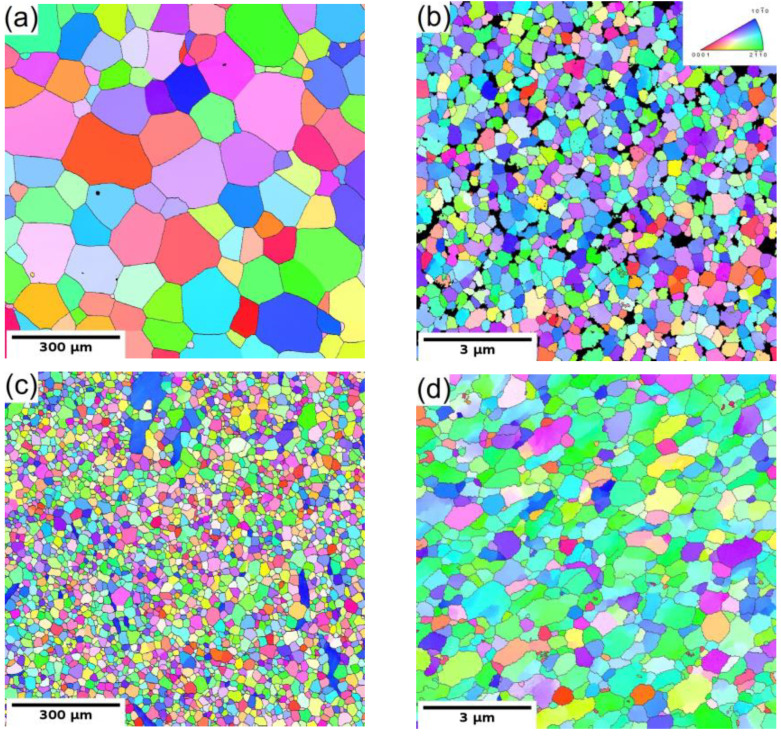
The EBSD/TKD orientation maps of both investigated alloys: (**a**) WE43-0P, (**b**) WE43-8P, (**c**) W3-0P, and (**d**) W3-8P sample.

**Figure 2 materials-15-07571-f002:**
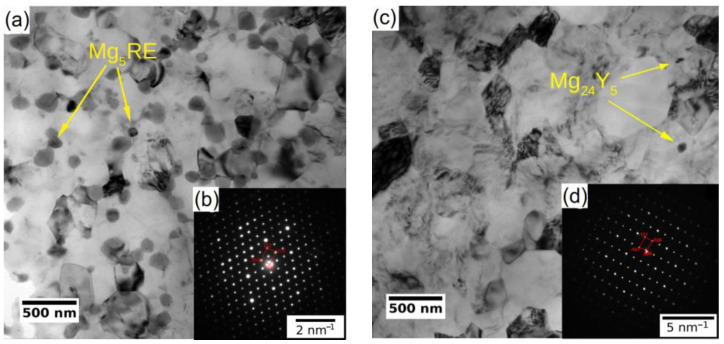
TEM micrographs (bright field) of UFG conditions of (**a**) WE43-8P and (**c**) W3-8P. Representative SAED patterns of (**b**) Mg_5_RE and (**d**) Mg_24_Y_5_ intermetallic particles.

**Figure 3 materials-15-07571-f003:**
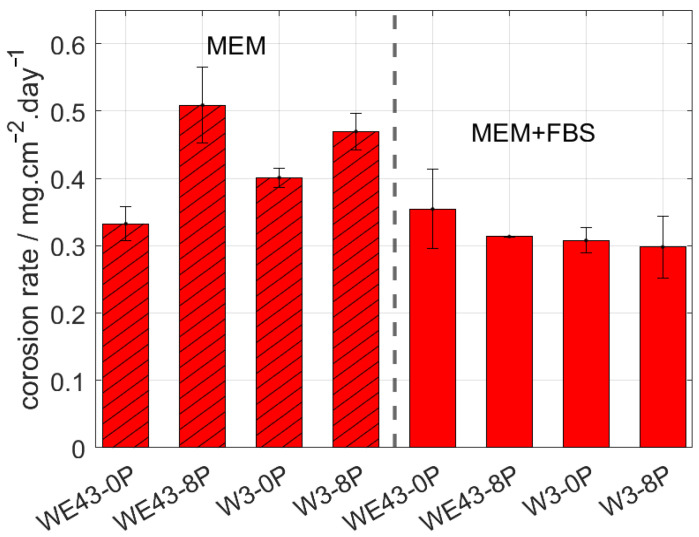
The corrosion rate of the investigated materials after 14 days of immersion in MEM and MEM + 10% FBS biological media.

**Figure 4 materials-15-07571-f004:**
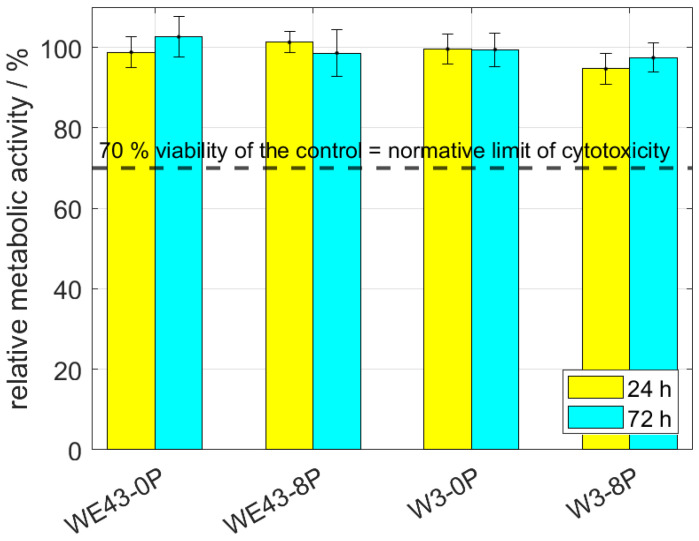
The relative metabolic activity after 24 h and 72 h of L929 murine fibroblasts—elution tests. The dashed line stands for the normative limit of 70% metabolic activity of the control. Error bars express the sample standard deviation of two-four samples measured in six iterations.

**Figure 5 materials-15-07571-f005:**
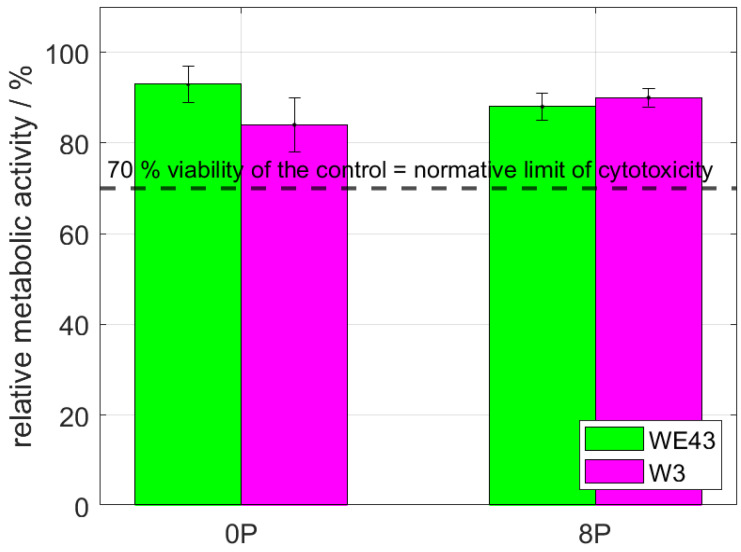
The relative metabolic activity after 24 h of L929 murine fibroblasts—direct tests. The dashed line stands for the normative limit of 70% metabolic activity of the control. Error bars express the sample standard deviation of two-four samples measured in six iterations.

**Figure 6 materials-15-07571-f006:**
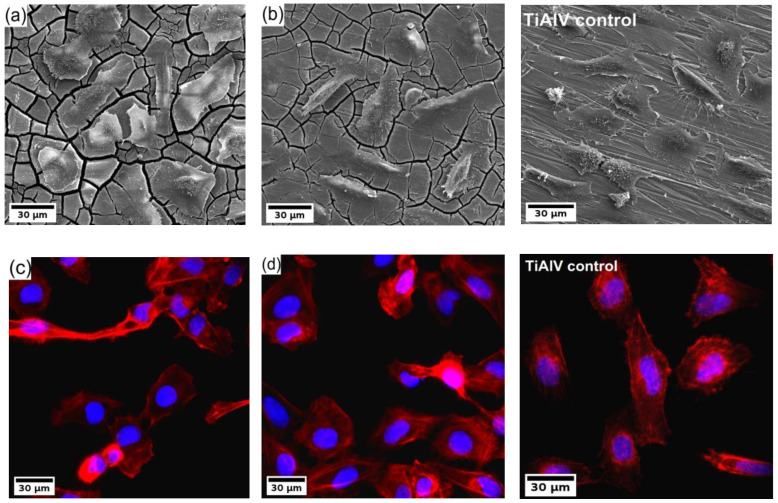
(**a**,**b**) SEM micrographs and (**c**,**d**) fluorescence images of U-2 OS human osteosarcoma cells (nuclei in blue, F-actin in red) on the WE43-8P (**a**,**c**) and W3-8P (**b**,**d**) and Ti-6Al-4V control sample.

**Figure 7 materials-15-07571-f007:**
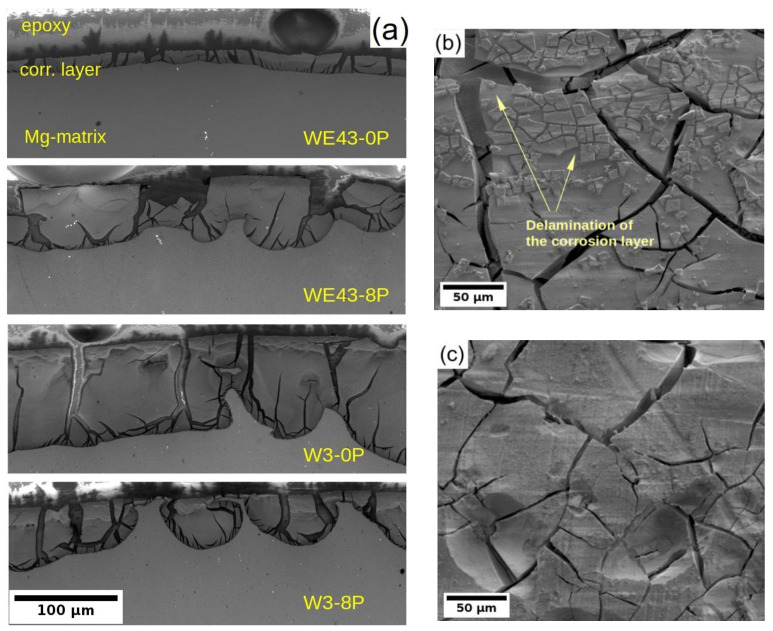
(**a**) the SEM micrographs of the corrosion layer cross-section formed on both investigated alloys/conditions and the surface of the (**b**) WE43-0P and (**c**) W3-0P samples. Measured after 14 days of immersion in the MEM biological media.

**Figure 8 materials-15-07571-f008:**
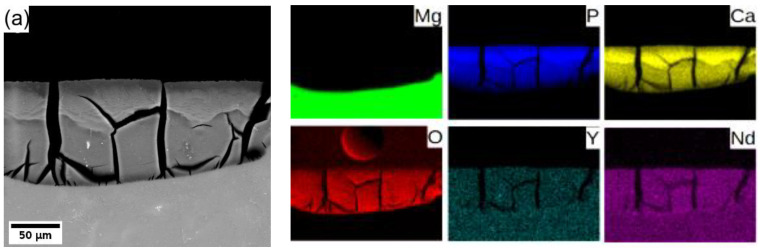
Elemental composition maps of the corrosion layer formed on (**a**) WE43-8P and (**b**) W3-8P after 14 days of immersion in MEM biological media.

**Figure 9 materials-15-07571-f009:**
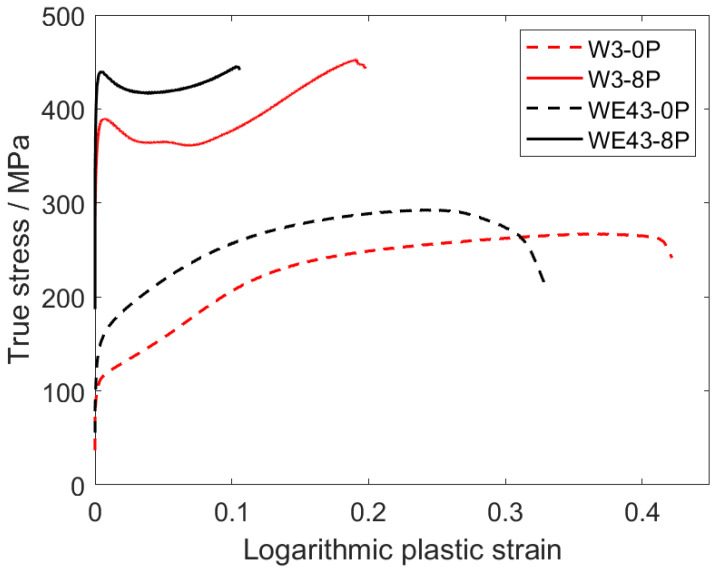
Compression deformation curves of the investigated samples.

**Figure 10 materials-15-07571-f010:**
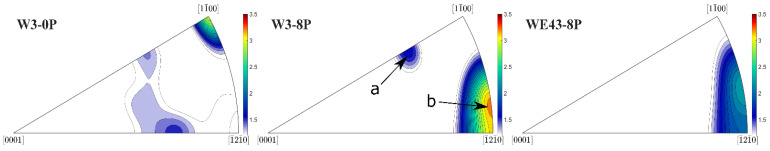
XRD inverse pole figures of the investigated samples calculated for the extrusion/ECAP processing direction.

**Table 1 materials-15-07571-t001:** The alloys composition measured by spark emission spectroscopy (wt%).

	Y	Nd	Gd	Dy	Zr	Fe	Cu	Ni	Mg
WE43	4.0	2.3	0.6	0.3	0.6	0.001	0.002	0.001	bal.
W3	3.1	-	-	-	-	0.03	0.004	0.0002	bal.

**Table 2 materials-15-07571-t002:** ECAP processing parameters.

	1P	2P	3P	4P	5P	6P	7P, 8P
WE43	335 °C 5 mm/min	315 °C 7 mm/min	300 °C 10 mm/min	295 °C 10 mm/min	290 °C 10 mm/min	285 °C 10 mm/min	285 °C 10 mm/min
W3	340 °C 3 mm/min	325 °C 3 mm/min	315 °C 5 mm/min	305 °C 7 mm/min	300 °C 7 mm/min	295 °C 7 mm/min	290 °C 7 mm/min

**Table 3 materials-15-07571-t003:** Results of the compression deformation tests and Vickers microhardness—HV_0.1_ of all the investigated samples.

	CYS/MPa	UCS/MPa	Deformation at Max. Stress	HV_0.1_
WE43-0P	125 ± 5	284 ± 10	0.26 ± 0.02	73 ± 5
WE43-8P	432 ± 3	451 ± 6	0.09 ± 0.02	114 ± 2
W3-0P	107 ± 3	265 ± 4	0.40 ± 0.02	57 ± 3
W3-8P	362 ± 6	448 ± 5	0.18 ± 0.01	100 ± 3

**Table 4 materials-15-07571-t004:** A literature review on mechanical properties reported for the Mg-Y-RE and Mg-Y alloys.

Materia/Treatment	CYS [MPa]	UCS [MPa]	Deformation to Fraction [%]	Grain Size [μm]	Refs.
WE43/Homogenization at 525 °C	150	220	10.2	70	[51]
WE43/Rotary swaging	285	415	7	0.61	[51]
WE43 /Multiaxial deformation 28P	210	300	17.2	0.93	[51]
WE43/ECAP 12P	260	300	12.4	0.73	[51]
WE43/SPS powder	230	430	23	1–10	[50]
WE43/Hot extrusion	250	-	-	20	[53]
WE43/3D-printed	208	395	-	-	[54]
Mg-3Y/Cast	90	230	57	40	[55]
Mg-0.6Y/ECAP 4P back pressure	140	280	23	0.4	[56]
Mg-8Y/Zone solidification	156	257	14	~500	[57]

## Data Availability

Not applicable.

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
