# Peer review of "Concurrence of High Corrosion Resistance and Strength with Excellent Ductility in Ultrafine-Grained Mg-3Y Alloy"

_materials, 2022, doi:10.3390/ma15217571_

Round 1

Reviewer 1 Report

Dear Authors,

Thank you for submitting your paper to Materials. Your article presents the influence of omitting mischmetal and zirconium in magnesium alloy on investigated properties. The results suggest that applying the new alloy in biodegradable medical applications is possible. 

Your paper is clear to me and written in excellent English. In my opinion, it is almost ready for publication however there should be always some questions so can you explain why when listing the dimensions of the specimen you use the cubic meters as a unit? These lengths describe volume but they should be defined in units of length.

Figures 2c and 2d are hardly readable

The first sentence of section 3.5 is not good. You did not test the mechanical properties of "investigated conditions" but the properties of materials.

What is your definition of plastic deformation to fracture since it seems that data in Tab. 4 do not comply with Fig. 10?

You presented average hardness but what was the data deviation?

Best regards,

Reviewer 2 Report

The manuscript has been well organized and written. Before acceptance, the author needs to polish the paper. 

1. Please use wt%.

2. Do not use the abbreviated words in the abstract and conclusion. 

3. Please supply more details in the introduction why the author selected Mg-3Y alloy.

4. The details of facilities should be added, such as corporation, city, country.

5. which kind of RE metal element exists in the alloy?

6. Please supply the XRD patterns of Mg alloys. And if possible, to calculate the phase compositions using the Rietveld method.

7. Could the Lotgering factor be calculated to characterize the texture microstructure?
